# Consumer Willingness to Recycle The Wasted Batteries of Electric Vehicles in the Era of Circular Economy

Miaomei Guo and Weilun Huang *

School of Finance and Trade, Wenzhou Business College, Wenzhou 325015, China
* Correspondence: huangwl@wzbc.edu.cn

**Abstract:** Electric vehicles (EVs) are increasingly being used for the benefit of the environment and to foster the development of a low-carbon circular economy. However, compared to internal combustion engine cars, spent EV batteries (WBEVs) constitute a different form of waste, and their recycling mechanism is still in its early stages. WBEV consumer willingness to recycle is an issue in a circular economy in which EV users should be WBEV recycling pioneers. The purpose of this article is to develop an analytical model for consumers' desire to return WBEVs for recycling, based on the circular economy and consumer welfare, in order to investigate consumer incentives for the construction of a WBEV recycling system. PLS-SEM was used for the analysis, and the results revealed the following. First, both the perception of government policy and environmental attitudes have significant positive causal effects on consumers' intentions to recycle. Second, the perception of benefits has a significant positive mediating effect on recycling intention, whereas the perception of loss has a significant negative mediating effect. Third, the multigroup analysis found that, with the exception of gender, the variables of age, income, education, area of residence, recycling experiences, and EV ownership all have substantial moderating impacts, although their routes and directions vary considerably. Recycling policies must be appropriate for consumers, and this has policy consequences for the circular economy. Environmental education and incentives should be provided to increase consumer knowledge and willingness to recycle. Big data might help with the design of a WBEV recycling system. It is necessary to create an intelligent recycling platform, cross-regional recycling collaboration, and smart logistics for WBEVs. Further, the battery refill mechanism of energy replenishment might encourage the recycling of WBEVs.

**Keywords:** electric vehicles; wasted batteries; battery recycling; consumer willingness to recycle; circular economy

## 1. Introduction

Global warming and the global consensus on carbon neutrality have accelerated the global growth of the electric vehicle (EV) sector. However, recycling methods for EVs and spent batteries of electric cars (WBEVs) are still in the early stages, showing that WBEV recycling is mostly driven by regulation. Thus, the policy challenge of EVs' circular economy has been consumers' desire to recycle WBEVs. The EU has an increased producer responsibility, and nations such as China, Japan, and India have restrictions on WBEVs to promote a circular economy. However, most countries' WBEV recycling regulations are still insufficient. According to Rajaeifar et al. [1], remanufacturing, reusing, and recycling are the best treatments for WBEVs in terms of the circular economy.

Many governments have encouraged the use of electric vehicles, with China leading the way. By the end of March 2022, China's total number of pure electric cars (EVs) had reached 7.25 million, accounting for 2.36% of all vehicles, with an annual growth rate of more than 100%. In Europe, more than 1.4 million EVs will be registered in 2022, with a use policy objective of at least 30 million registered EVs by 2030. By 2020, the number of EV registrations in the United States will have surpassed one million. The US policy aim

for EV adoption is for EVs to account for 50% of new car sales by 2030. The EV sector is predicted to flourish, creating a strong need for EV batteries, and discarded EV batteries are a hazard to the environment. The issues are as follows. (1) An EV battery should be changed when its power capacity decreases to 70–80% of its original level [2]. (2) Replacing EV batteries will result in a rising waste stream of WBEVs in the future [3,4]. (3) WBEVs include dangerous metals such as Hg, Pb, and nickel [5] and can pollute soil, air, and subterranean water if improperly treated [6].

The fast expansion of the EV sector, along with a scarcity of resources, is driving up the price of lithium battery metals throughout the world, with the price of lithium carbonate growing from RMB 67,906/ton in early 2021 to RMB 589,000/ton in November 2022. WBEVs have a high concentration of battery-grade materials such as copper and organic electrolytes [7]. WBEV recycling is a notable alternative for battery producers to ease material scarcity and minimize material prices [6]. Tesla, Ultium Cells, and SK Innovation are among the companies that have begun to put out their plans for WBEV recycling.

More and more nations and regions have set goals for recycling WBEVs, since this has positive effects on the environment and the economy. The "Law for the Promotion of Effective Utilization of Resources" was introduced in Japan in 2000, and it stipulates that Li-ion rechargeable batteries must have a recovery rate of 30% or higher [8]. By 2020 and 2025, respectively, the General Office of the State Council of China has mandated a recovery rate of 40% and 50% for critical waste [9]. For new batteries to be manufactured after 2030, the European Commission has established a new guideline requiring battery manufacturers to satisfy a certain level of recycled materials [10].

China is the EV market's pioneer and the world's largest producer of EV batteries. WBEV recycling is extremely difficult and essential. The recycling market for WBEVs is expanding in China, with a growth rate of more than 50% expected in 2020 and 2021. To aid management, the Chinese government has enhanced recycling infrastructure and implemented the Extended Producer Responsibility (EPR) system as well as an EV battery traceability labeling system. Through public–private partnerships and third-party services, the government is also promoting private capital to engage in garbage recycling [9]. By 15 April 2022, 14,967 recycling outlets had been established in China by EV manufacturers, as well as cascade utilization firms such as GAC Mitsubishi and BAIC New Energy, mostly in the Bei-jing-Tianjin-Hebei, Yangtze River Delta, Pearl River Delta, and central regions.

The recycling rate of WBEVs, on the other hand, is not proportionate to EV penetration in the vehicle consumer market [11]. On 31 July 2018, China's complete management platform for power battery recycling and traceability was officially launched. Power batteries built prior to the system's implementation are not subject to the system, and users who purchased EVs prior to the system's implementation have additional options for disposing of WBEVs. When the battery power capacity falls below the recommended value, EV owners have at least three options for disposing of the discarded batteries: (1) recycle the WBEVs at automotive sales service stores (4S stores), battery leasing companies, or recyclers; (2) leave the end-of-life (EOL) EVs or WBEVs at home; or (3) dispose of the EOL EVs or WBEVs in the wilderness. Although it may appear nonsensical to dump EOL EVs or WBEVs, EVs are frequently reported abandoned in the woods. In Jiangsu and Zhejiang provinces, respectively, twenty EV buses and 4000 shared EVs were abandoned in the wilderness in 2019. Over 3000 shared EVs were discovered in an abandoned parking lot in Guangdong province in 2020. Nearly 2000 EVs were reported abandoned in Hangzhou, Zhejiang province, in 2021. When it comes to recycling WBEVs, consumers encounter numerous challenges, including: (1) the expensive cost of battery replacement; (2) a lack of understanding as to how to recycle; (3) restricted access to recyclers; (4) limited recycling outlets; and (5) a lack of subsidies or other recompense from the government or recyclers [12].

Consumers play an important role in the recycling industry [12]. According to Long et al. [13], consumers are the starting point for the recycling chain. Consumers return garbage to recyclers in order for the waste to be recycled. The recycling rate is determined by their pro-environmental attitude and awareness. Kumar [14] claimed that one of the



most important reasons for increased waste is consumers' inability to return unwanted items to producers. Dhir et al. [15] also asserted that the e-waste situation was caused by consumers' poor recycling participation, and that better knowledge of consumers' recycling intentions is desperately required to urge them to recycle. Consumers own EVs, and are the principal suppliers of WBEVs. Understanding the recycling intentions of consumers or the reasons for their readiness to return WBEVs is critical to reducing WBEV waste. Further, consumers have not been given enough consideration in the existing literature. As a result, the purpose of this article is to bridge this gap by investigating the factors that influence consumers' propensity to return WBEVs for recycling.

The theoretical contributions of this paper are as follows: (1) an expansion of behavioral economics and environmental economics by exploring consumers' willingness to recycle (WTR) WBEVs; (2) an expansion of the theory of wasted battery recycling, as this paper is one of the first works to discuss the recycling of WBEVs from the perspective of the consumer; and (3) a reference for waste management research with the structure and method of analysis of the paper.

In recent years, most governments and economies, including China and the United States, have proposed the idea of the circular economy as a fundamental premise for industrial development [16,17]. The goal of a circular economy is to keep resources in the economy for as long as possible through recycling or reusing garbage and its byproducts, as well as prolonging the service life of products [18]. Without appropriate waste management, a circular economy cannot be realized [19]. According to Ranjbari et al. [19], substantial research has been conducted on waste management within the circular economy domain during the previous two decades, with 962 relevant publications published in over 200 journals.

These articles can be categorized into seven major themes: circular economy transition, environmental impacts and lifecycle assessment, bio-based waste management, electronic waste, municipal solid waste, plastic waste, and building and demolition waste. However, the management of WBEVs is omitted, showing a lack of study on WBEV management in the context of the circular economy. Recycling is a component of waste management, and this study investigates consumers' recycling intentions for WBEVs, contributing theoretically and practically to waste management and the creation of a circular economy in the following areas: (1) an analytical framework for the study of consumers' recycling intentions for WBEVs is provided; (2) the influencing factors of consumers' WBEV recycling behavior are clarified and discussed from the three perspectives of causality effect, mediating effect, and mediation effect; and (3) incentive measures and policies for government and recyclers are proposed to encourage consumers' recycling intentions for WBEVs, based on the key influencing factors.

The Section 2 of this study uses the findings of a literature review and an expert survey to define the theoretical framework and structural equation model in this research, and then our hypotheses are defined. The Section 3 examines the statistical results for consumers' WTR for WBEVs and the elements that influence these consumers, such as causal effects, mediator effects, and moderator effects. The Section 4 includes empirical hypothesis results and policy implications. The Section 5 further includes the conclusion, limitations, recommendations, and future research prospects.

## 2. Literature Review and Hypotheses

The circular economy should be a market economy that allocates macro resources to ensure its success. Thus, in the age of the circular economy, consumers' WTR for WBEVs is a crucial policy problem. The essential concept of the circular economy is resource recycling, which should be feasible in all aspects of the national economic reproduction system (including consumption and use). Natural resources should be used rationally in the production and processing process, energy and raw materials should be processed into environmentally friendly products as much as possible using advanced technology and onsite recycling, and final products should be consumed in a rational manner during the

circulation process and consumption. These desiderata originate in resource recovery in the manufacturing and processing processes.

WBEV recycling is receiving more attention, and the relevant research is expanding. Figure 1a demonstrates that papers on WBEV recycling were uncommon prior to 2016, but their numbers began to increase after 2017. However, this report found only 851 articles with the keywords "EV battery recycling," "electric vehicle recycling," or "power battery recycling" in the web of science (WOS) core collection database on 17 September 2022. Similarly, as shown in Figure 1b, on 17 September 2022, this paper found only 695 articles containing the keywords "car battery recycling" or "power battery recycling" on the China National Knowledge Infrastructure (CNKI) website.

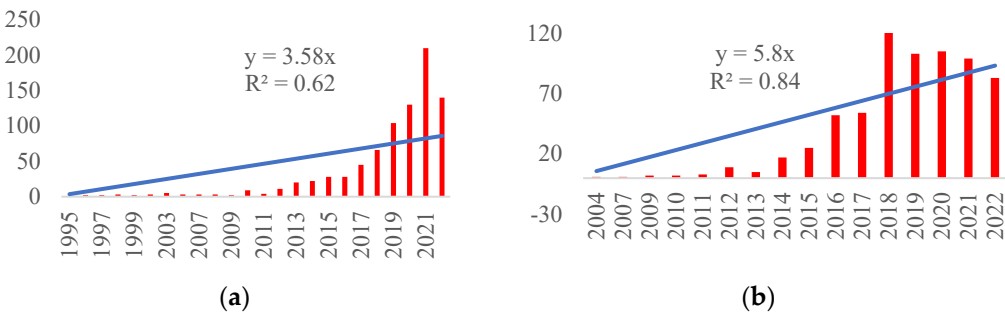

**Figure 1.** Articles with the topic of WBEV recycling in WOS and CNKI: (**a**) articles with the topic of WBEV recycling from 1995~2022 in WOS; (**b**) articles with the topic of WBEV recycling from 2004~2022 in CNKI.

### 2.1. Recycling Modes of WBEVs

The literature primarily discusses four types of recycling systems: (1) a recycling mode driven by manufacturers ($Mode_M$), who recycle WBEVs directly from consumers through their sales channels [20,21]; (2) a recycling mode driven by retailers ($Mode_R$), whom the manufacturers have encouraged to take on the responsibility for recycling [4]; (3) a recycling mode ($Mode_{TP}$) in which producers outsource the recycling process to well-equipped third parties [21,22]; (4) a recycling mode ($Mode_A$), that is often constituted of manufacturers or sellers and has the benefit of a scale economy [21].

The ideal recycling structure and pathways have been addressed in the literature. Tang et al. [4] investigated the social welfare implications of single-channel recycling modes (as $Mode_M$, $Mode_R$, and $Mode_{TP}$) as well as competing dual-channel recycling modes (as mode $Mode_{M,R}$, $Mode_{M,TP}$, and $Mode_{R,TP}$), concluding that $Mode_{M,R}$ is the optimal recycling method. According to Zan and Zhang [21], the ideal recycling method for manufacturers is determined by the trade-offs between recycling and reuse costs. Li [22] examined the EV battery supply chain using $Mode_M$, $Mode_R$, $Mode_{TP}$, and dual-channel modes, claiming that the best mode to use is determined by competition and third-party economies of scale.

WBEV recycling is mostly carried out by industry organizations and partnerships in Europe and the United States, and the government has developed a deposit system to encourage battery registration and required restitution [23]. In Japan, battery manufacturers create recycling routes using reverse logistics, and recycling is subsidized by the government [23]. According to the "Measures for the Management of Recovery and Utilization of Power Batteries for New Energy Vehicles" law in China, EV producers are responsible for recycling WBEVs and must develop power battery recycling routes within the EPR system. Furthermore, co-construction encourages the creation and sharing of recycling channels among EV makers, battery producers, automobile scrapping facilities, dismantling businesses, and comprehensive utilization companies. However, China's WBEV recycling business is still in its early stages. The WBEV recycling market has been dominated by small- and medium-sized recycling businesses [24], resulting in a disorderly and informal recycling industry.

### 2.2. Literature Review on Consumers' WTR WBEVs

Consumers are the most important participants in the WBEV recycling process. Since the Chinese government encouraged the use of EVs in 2015, the market share of EVs has grown year after year. EV market share has increased from 0.34% in 2015 to 2.6% in 2021, as shown in Figure 2a. EV batteries have a lifespan of 5–8 years. Consumers can change batteries when their performance is no longer good. The retired batteries form a massive mound of WBEVs. Typically, consumers have four options for participating in WBEV recycling:

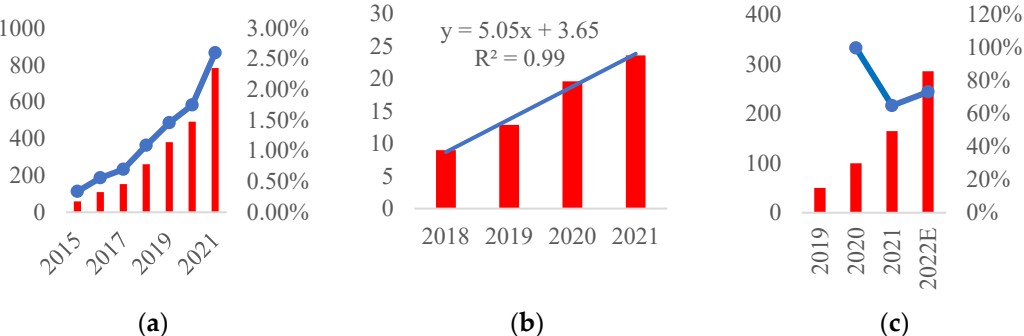

**Figure 2.** (**a**) Statistics of EV ownership in China from 2015 to 2021 (10,000 pieces). Data source: China Economic Industry Research Institute, Ministry of Public Security. Note: The percentage illustrates the proportion of EV ownership in the total car ownership. (**b**) Recovery volume of lithium batteries in China from 2018 to 2021 (10,000 tons). Data source: EV Tank, China Economic Industry Research Institute. (**c**) Recycling market size of power lithium batteries in China from 2019–2022 (RMB 100 million). Data source: EV Tank, China Economic Industry Research Institute. Note: the percentage indicates the annual growth rate; 2022E is a prediction.

(1) Return the EOL EVs to 4S stores or auto salvage yards.

(2) Return the WBEVs to 4S stores or battery leasing businesses.

(3) Return the WBEVs to recycling facilities. Collected WBEVs will be transported to recycling centers via 4S retailers, vehicle salvage yards, battery leasing firms, or directly by customers. Following this, WBEVs will be cascade used or recycled (the flow of WBEVs is illustrated in Figure 3). Recycling centers are official recyclers that are primarily made up of EV manufacturers and cascade usage firms. Formal recyclers are regulated and approved facilities that process garbage with some level of industrial hygiene and worker protection, and they are typically equipped with the finest recycling equipment possible and are subject to highly stringent environmental protection standards [25–27]. With the expansion of the recycling sector, the number of batteries recycled from official recyclers is quickly growing. As seen in Figure 2b,c, the recycling volume of lithium batteries in China rose by more than 20% between 2020 and 2021, and the recycling market scale expanded by more than 50% over the same period.

(4) Informal recyclers buy the WBEVs or EOL EVs. Unlike formal recyclers, informal recyclers are not regulated by the government [25,28]. They are tiny and disadvantaged businesses with a low profit threshold, no registration, no tax, no social welfare benefits, and a high labor intensity [29,30]. Informal recyclers have fewer operational expenses and can offer greater prices than authorized recyclers. Despite the advantages of authorized recyclers, most consumers choose to sell WBEVs to informal recyclers, since they often provide substantial compensation and door-to-door service [31]. On the other hand, informal recycling is harmful to the environment and endangers human health.

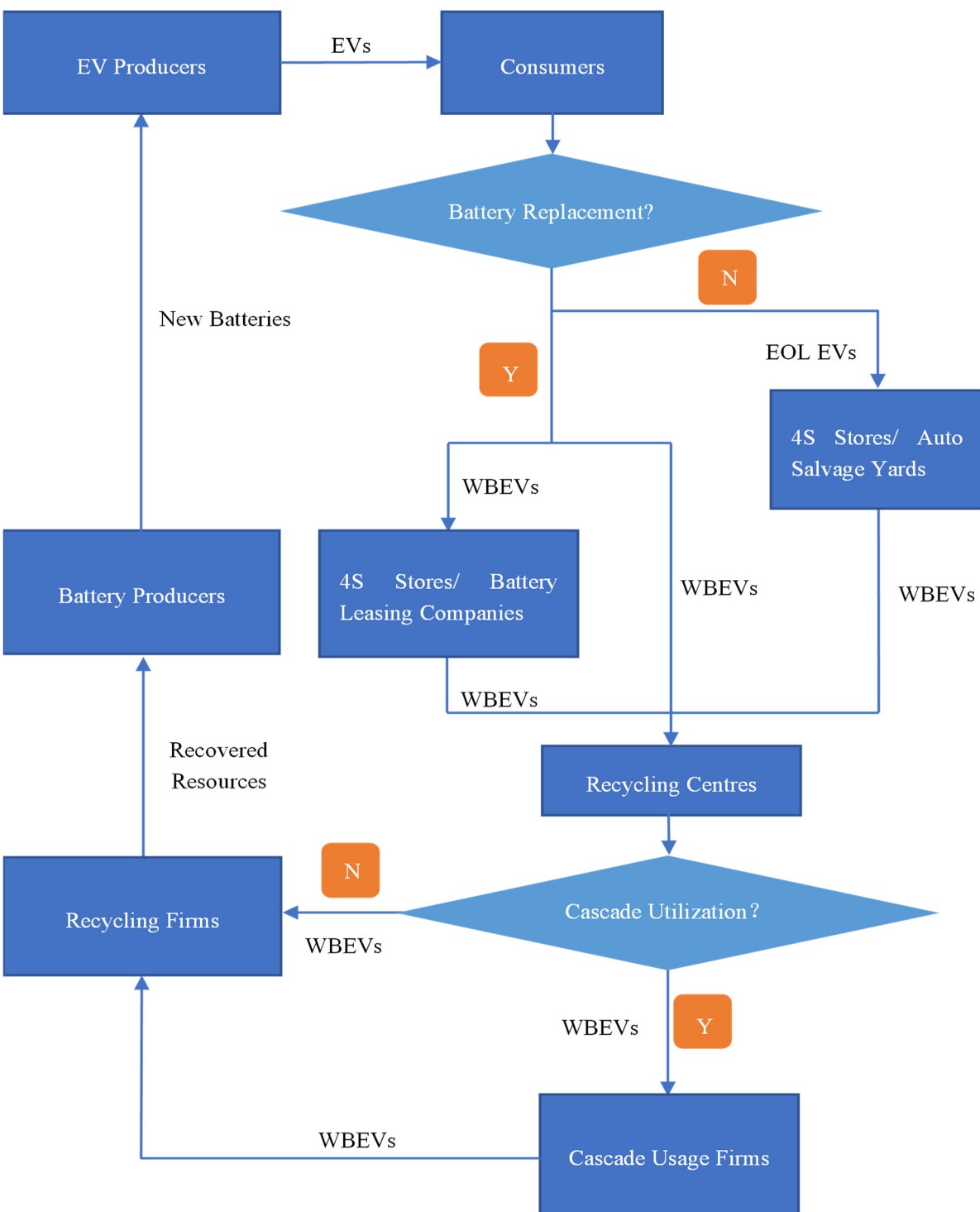

**Figure 3.** Recycling structure and the flow of WBEVs.

Consumers have important responsibilities in the recycling industry. Chen and Tung [32] have stated that the management of the recycling program would be impossible without the involvement of the government and consumers. Consumers are the first link in the recycling cycle [33]. According to Gaur and Mani [34], consumers are providers of returned items and are a significant factor in re-manufacturing continuity. Budijati et al. [35] further noted that consumers play the role of suppliers in the take-back program, and their intention to return old items has a substantial influence on the program's efficacy. Understanding consumers' WTR is critical for the long-term development of the recycling

sector [35]. Consumer engagement is critical to increasing recycling rates [36]. Recycling efficiency is heavily dependent on the consumer's recycling knowledge [37]. Sarath et al. [37] proposed that the low recycling rate in both developing and developed countries was due to a lack of information about recycling among the general public.

The aims of consumers when recycling waste such as e-waste, municipal solid trash, and domestic garbage have frequently been studied in the literature. In the study of consumers' recycling intentions, theories of consumer behavior, such as the theory of planned behavior (TPB), the theory of reasoned action, the technology acceptance model, and normative activation theory, are commonly used [38–49]. TPB is most commonly utilized to comprehend human intentions and behavior [39]. To examine the effects of psychological variables on consumers' recycling intentions, subjective norms, convenience, awareness, attitudes, perceived behavioral control, and moral norms can be added into the TPB model. Previous research has found that perceived benefits [15], environmental knowledge [39], a good attitude toward the environment [50], convenience [51], information security [52], and nostalgia [53] can all influence consumers' recycling intentions and behaviors.

The willingness of consumers to recycle WBEVs is a key aspect in the circular economy sector. Aside from consumer behavior theories, the following ideas can help us to understand customers' propensity to recycle WBEVs. (1) Circular economy theories, such as closed-loop supply chains and reverse logistics, demonstrate the relevance of consumer recycling intentions in supporting a circular economy. (2) Waste management theories, such as the 3Rs (reduce, reuse, and recycle), illustrate consumers' roles in establishing a sustainable waste management system. (3) Social network theories examine how customers' recycling intentions are impacted by their social networks. (4) Behavioral economics theories, such as the theory of nudge, help to evaluate the ways in which little changes in the design of recycling programs might influence customer recycling intentions. (5) Environmental psychology theories, such as the idea of pro-environmental behavior, exemplify the ways in which environmental knowledge and concerns influence consumer recycling intentions.

Investigation into the recycling of WBEVs has barely begun. The majority of relevant research has focused on analyzing vehicle manufacturers, battery manufacturing companies, automotive retailers, and recycling enterprises, but it overlooks the role of consumers and seldom considers approaches that encourage consumers to participate in WBEV recycling [12]. Flygansvaer et al. [54] underlined the importance of researching and understanding consumers. There is little literature on consumers' readiness to recycle WBEVs. On 18 October 2022, the authors searched the CNKI website for the terms "recycling behavior of EV battery" or "recycling aim of EV battery" and found no related publications.

Similarly, this report uncovered just one publication using the keywords "recycling behavior of EV battery" or "recycling aim of EV battery" in the Web of Science (WOS) core collection database. Dong and Ge [12] investigated the factors that influence consumers' intention to recycle WBEVs in China and discovered that factors such as perceived behavioral control and subjective norms have significant impacts on consumers' WTR WBEVs, and demographic variables such as age, EV ownership, and regional groups have significant moderating effects. Some studies have also looked into consumers' intentions to recycle WBEVs. Zhang et al. [46] investigated the elements that influence consumers' desire to recycle lead-acid waste batteries and discovered that consumers' performance expectancy, social influence, and facilitating environments can improve their willingness to recycle. The anticipation of effort, on the other hand, has a negative effect. Tang et al. [4] used an online poll to assess consumers' WBEV recycling knowledge and intentions and found that 89.86% believed it was vital to recycle, but 56.9% understood little about the recycling process. Furthermore, they found that recycling ease, pricing, and environmentally friendly disposal are important elements influencing consumers' WTR, and 21.9% of respondents would evaluate whether recycling routes are official.

### 2.3. Variables, Study Architecture, and Hypotheses

The dependent variable in this article is consumers' willingness to recycle ($WTR_0$), as determined by the findings of the expert interviews and the literature study. The efficacy of government policy as seen by the public ($GP_0$) and consumers' attitudes toward the environment ($EA_0$) are two independent factors that affect the motivation of $WTR_0$. Consumers' perceptions of benefits ($PB_0$) and perceptions of loss ($PL_0$) are the two mediating factors. Additionally, moderators for demographic factors have been specified. Figure 4 provides an illustration of the study's structure and hypotheses.

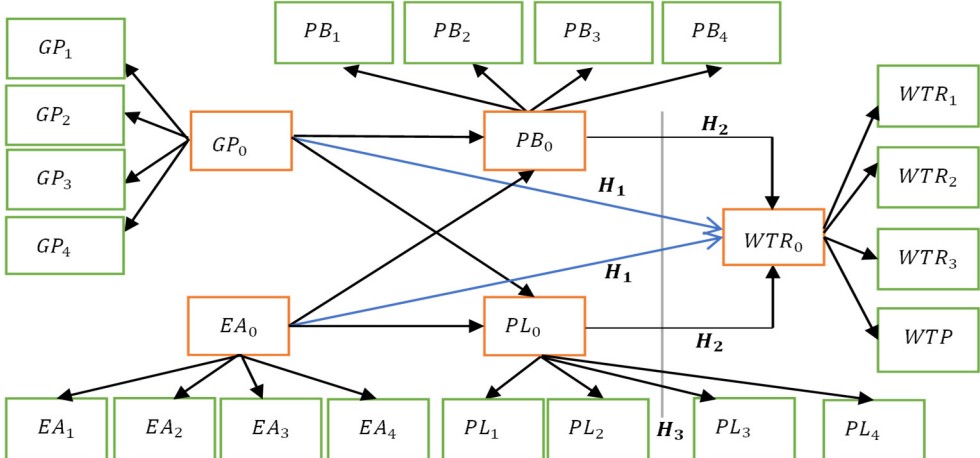

**Figure 4.** Study architecture and hypotheses.

Consumers' recycling strategies can be categorized as "participate" or "not participate," and their WTR may vary depending on the economic results. Therefore, three alternative situations are used to measure "$WTR_0$": (1) $WTR_1$: consumers' WTR for a recycling fee; (2) $WTR_2$: consumers' WTR for a recycling reimbursement; (3) $WTR_3$: consumers' WTR for neither a reimbursement nor a recycling fee. Additionally, consumers' willingness to pay (WTP) is offered as a financial assessment for consumers' WTR [55,56].

The first motivator of $WTR_0$, $GP_0$, is judged by consumers' perceptions of policy efficacy in boosting recycling market growth, improving recycling supervision, warning of unlawful recycling behavior, and restraining residents' recycling behavior. Recycling legislation and regulations limit consumers' recycling behavior. However, its impact is determined by the government's capacity to implement policy and consumers' perceptions of the success of government policy.

Asymmetric information theory holds that information disclosure is critical in determining consumer trust in the government. Similarly, according to source credibility theory, message exposure is the most important element in determining families' faith in the government [57]. $GP_0$ is measured in this article by perceptions of policy efficacy in the following areas: (1) $GP_1$: efficacy in promoting WBEV recycling; (2) $GP_2$: efficacy in supervising WBEV recycling; (3) $GP_3$: efficacy in alerting against illegal recycling; and (4) $GP_4$: efficacy in standardizing recycling disposal behavior.

The second motivator of $WTR_0$, $EA_0$, pertains to consumer attitudes about the environment. $EA_0$ is determined by consumers' assessments of environmental effects in the following scenarios: (1) $EA_1$: irresponsible disposal of WBEVs; (2) $EA_2$: incineration of WBEV components; (3) $EA_3$: random dumping of WBEVs; and (4) $EA_4$: inappropriate disassembly of WBEVs. Some research has suggested a substantial positive association between environmental views and recycling behavior [58]. Kurz et al. [59] discovered a minimal link between environmental concern and recycling behavior.

According to the consumer theory of microeconomics, consumers make decisions based on the utility maximization principle [60]. On the other hand, consumers are frequently reported to stray from utility maximization theory, because traditional consumer

theory is driven by the market mechanism, and it believes that consumers are completely rational and market knowledge is comprehensive [61]. These assumptions are frequently contradicted by reality. As a result, behavioral economics incorporates psychology into classical economics to explain the disparity [62]. The cost–benefit analysis technique, which is based on behavioral economics, is useful in analyzing recycling-related decision-making behavior, such as the recycling of resource materials [63], the construction of reverse logistics systems [64], the recycling of onsite residential graywater [65], the recycling of e-waste [66], the recycling of waste photovoltaic modules [67], and the construction of a municipal solid waste recycling facility [55].

On the basis of cost–benefit theory, this article examines the mediating effects from an economic standpoint. As mediators, $PB_0$ and $PL_0$ are presented. $PB_0$ refers to consumers' understanding of the positive effects of WBEV recycling on the environment, society, or individuals. The purported benefits of recycling include decreasing trash disposal in landfills, conserving resources, lessening negative environmental consequences, generating economic benefits, and educating youngsters [68]. Similarly, $PB_0$ is determined by consumers' assessments of the benefits of WBEV recycling, including: (1) $PB_1$: resource utilization optimization; (2) $PB_2$: risk reduction of environmental contamination; (3) $PB_3$: land saving; and (4) $PB_4$: economic benefits. Previous research has shown that perceived benefits influence residents' pro-environmental intentions and behaviors [15,69,70].

$PL_0$ denotes the monetary or non-monetary expenses involved in the recycling of WBEVs. A lack of acceptable facilities, a complicated operational method, time consumption, great distance, and economic loss are all factors that contribute to perceived loss [68,70]. Knussen et al. [71] discovered that consumers' inability to recycle was due to a lack of facilities. Chen and Tung [32] have shown that the consumer impression of a lack of facilities has a substantial impact on their intention to recycle. In this article, $PL_0$ is defined as consumers' assessment of the loss caused by WBEV recycling, including: (1) $PL_1$: difficulty in locating a recycler; (2) $PL_2$: annoyance caused by a shortage of recycling outlets; (3) $PL_3$: inconvenience caused by a great distance; and (4) $PL_4$: the time consumption of the operation.

Based on the above study, the model's routes (hypotheses) are as follows:

$$WTR_0 = \alpha_1 + \beta_1 GP_0 + \varepsilon_1 \tag{1}$$

$$WTR_0 = \alpha_2 + \beta_2 EA_0 + \varepsilon_2 \tag{2}$$

$$WTR_0 = \alpha_3 + \beta_3 GP_0 + \beta_4 EA_0 + \varepsilon_3 \tag{3}$$

$$WTR_0 = \alpha_4 + \beta_5 PB_0 + \beta_6 PL_0 + \varepsilon_4 \tag{4}$$

$$PB_0 = \alpha_5 + \beta_7 GP_0 + \beta_8 EA_0 + \varepsilon_5 \tag{5}$$

$$PL_0 = \alpha_6 + \beta_9 GP_0 + \beta_{10} EA_0 + \varepsilon_6 \tag{6}$$

$$WTR_0 = \alpha_7 + \beta_{11} GP_0 + \beta_{12} EA_0 + \beta_{13} PB_0 + \beta_{14} PL_0 + \varepsilon_7 \tag{7}$$

where the residual terms are $\varepsilon_1, \varepsilon_2, \varepsilon_3, \varepsilon_4, \varepsilon_5, \varepsilon_6, \varepsilon_7$. Equations (1)–(3) test the hypothesis $H_1$, that $GP_0$ and $EA_0$ have a considerable influence on $WTR_0$ $\left( \begin{matrix} GP_0 \\ EA_0 \end{matrix} \rightarrow WTR_0 \right)$. Equations (4)–(7) test the alternative hypothesis $H_2$, that mediators have substantial mediating effects on the connections between the independent and dependent variables $\left( \begin{matrix} GP_0 \\ EA_0 \end{matrix} \rightarrow \begin{matrix} PB_0 \\ PL_0 \end{matrix} \rightarrow WTR_0 \right)$. Furthermore, $H_3$ assumes that the direct and mediated impacts are somewhat varied depending on the characteristics of the consumers, namely, gender, age, income, education, place of residence, and recycling experience. The following theories are based on the preceding discussion:

**$H_1$**. There are significant causal relationships between consumers' WTR and motivations $\left( \begin{matrix} GP_0 \\ EA_0 \end{matrix} \rightarrow WTR_0 \right)$.

$H_2$. There are significant mediator effects regarding consumers' perceptions of benefits and perceptions of loss on $H_1 \begin{pmatrix} GP_0 & \rightarrow & PB_0 \\ EA_0 & \rightarrow & PL_0 \end{pmatrix} \rightarrow WTR_0$.

$H_3$. There are significant moderator effects of consumer's demographic variables such as gender, age, income, education, place of residence, recycling experience, and EV ownership on $H_1$ and $H_2$.

## 3. The Statistical Results for Consumers' WTR for WBEVs and Influencing Factors

### 3.1. PLS-SEM Modeling

The statistical method of partial least squares structural equation modeling (PLS-SEM) has been employed frequently in the literature on consumers' recycling intentions and behaviors [72,73]. This study investigated the direct and indirect link between the important parameters indicated and consumers' WTR WBEVs, and the PLS-SEM technique was suited for the analysis. Structural equation modeling (SEM) is a type of second-generation multivariate analysis and an extension of standard linear modeling approaches [74,75]. Researchers can use SEM to: (1) investigate correlations between unobservable variables [75], (2) assess models with numerous constructs and latent and observed variables [74], and (3) evaluate statistical associations concurrently [76]. Variance-based PLS-SEM and covariance-based SEM (CB-SEM) are two statistical approaches for SEM [76–78]. CB-SEM is more suited to theory testing or confirmation, whereas PLS-SEM is better suited to prediction and theory building [77]. PLS-SEM is superior to CM-SEM in the following scenarios: (1) the theoretical models are complicated, with numerous indicators and constructs; (2) the sample size is limited; (3) the data distribution is non-normal; and (4) strong assumptions cannot be fully satisfied [77,78].

### 3.2. Descriptive Statistics

On February 2022, an online survey was distributed to Chinese EV buyers. A total of 335 valid replies were received. The data acquired met Chin's [79] minimum sample size recommendation of ten times the most significant number of independent latent variables. The following are the descriptive statistics of the respondents' characteristics: (1) In terms of gender ($P_G$), the sample included 183 males ($P_{G,M}$, 54.6%) and 152 females ($P_{G,F}$, 45.4%). (2) In terms of age ($P_A$), the median age was 31 years. (3) In terms of income ($P_I$), RMB 164,800 was the average family income. (4) In terms of education ($P_E$), 33.73% of respondents had a college degree or more, and the rest had less education. (5) In terms of residency ($P_R$), 60% of respondents dwelled in cities, and 40% were in rural areas. (6) In terms of recycling experience ($P_{RE}$), 73.4% of those polled had no prior recycling experience. (7) In terms of EV ownership ($P_O$), 83.8% of respondents owned or have owned an EV, either personally or through family members.

Consumers' WTR for WBEVs is statistically described as follows: (1) $WTR_1$: only 17.31% were willing to recycle when recycling is charged. (2) $WTR_2$: 62.39% were willing to recycle when they are rewarded for it. (3) $WTR_3$: 26.57% were willing to recycle without charge or reimbursement. These data suggest that charging for recycling (reimbursement) will raise recycling costs (benefits) while hindering (promoting) consumers' WTR, which is consistent with the findings of Escario et al. [58] that people's WTR is determined by the trade-off between predicted costs and benefits. (4) WTP: Consumers would prefer to spend less than RMB 100 for the recycling of WBEVs.

### 3.3. The Structural Equation Model

SmartPLS 3.0 was utilized in this work to estimate the path coefficient of the PLS-SEM model. The outcomes are depicted in Figure 5. Except for those between $GP_0 \rightarrow PL_0$, $EA_0 \rightarrow PL_0$, and $PL_0 \rightarrow WTR_0$, most path coefficients are positive. This is consistent with past research and our own experience. (1) The negative path coefficient between $GP_0 \rightarrow PL_0$ indicates that consumers with a higher perception of government policy effi-

cacy have a lower perception of loss. (2) The negative path coefficient between $EA_0 \rightarrow PL_0$ indicates that consumers with higher environmental attitudes have a lower perception of loss. (3) The negative path coefficient between $PL_0 \rightarrow WTR_0$ indicates that consumers with a higher sense of loss are less likely to recycle.

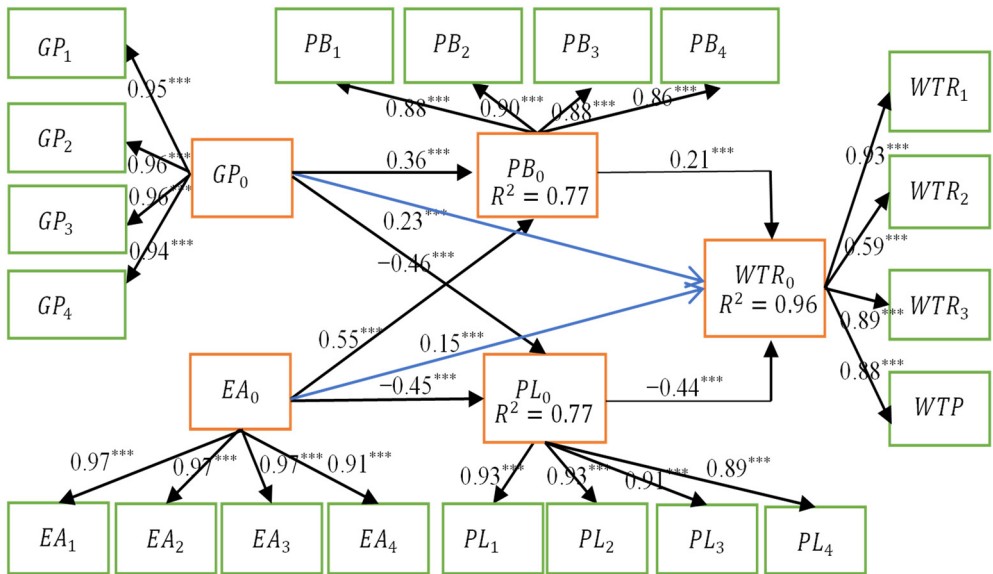

**Figure 5.** The path weighting scheme and the path coefficients of the PLS-SEM model. Note: *** $p < 1\%$.

The test of the model validity is illustrated in the Table 1. The indicators used to assess the reliability and validity of PLS-SEM model are Cronbach's alpha (*CA*), Dillon–Goldstein's rho (*rho_A*), composite reliability (*CR*), average variance extracted (*AVE*), $R^2$, and *Adj.R²*. According to Urbach et al. [80], *CA* and *CR* should not be lower than 0.6, and the proposed threshold value of *AVE* is 0.5. As shown in Table 1, the reliability and convergent validity of the variables are acceptable. In addition, bootstrapping was used to test the significance of path coefficients, and the results are shown in Table 2. According to the findings, $H_1$ and $H_2$ are well supported. Both $EA_0$ and $GP_0$ have large positive direct causal impacts on $WTR_0$. Furthermore, $PB_0$ and $PL_0$ strongly influence the link between $GP_0 \rightarrow WTR_0$ and $EA_0 \rightarrow WTR_0$.

The multi-group analysis approach was used to investigate the moderating impact of consumer attributes. Table 3 shows the results of moderating effects of $P_A$, $P_I$, $P_E$, $P_R$, $P_{RE}$, and $P_O$ for consumers whose gender had no significant moderating effects on their WTR. All moderators had no significant moderating effects on $EA_0 \rightarrow PB_0$, $PB_0 \rightarrow WTR_0$ and $EA_0 \rightarrow PB_0 \rightarrow WTR_0$, indicating that the impacts of $EA_0 \rightarrow PB_0$, $PB_0 \rightarrow WTR_0$ and $EA_0 \rightarrow PB_0 \rightarrow WTR_0$ were not significantly different among various groups of consumers.

**Table 1.** Reliability and convergent validity of the model.

| Variables | CA | rho_A | CR | AVE | $R^2$ | Adj. $R^2$ |
|-----------|------|-------|------|------|-------|------------|
| $WTR_0$ | 0.84 | 0.87 | 0.90 | 0.69 | 0.96 | 0.95 |
| $PB_0$ | 0.93 | 0.94 | 0.95 | 0.83 | 0.77 | 0.76 |
| $PL_0$ | 0.90 | 0.90 | 0.93 | 0.77 | 0.68 | 0.76 |
| $GP_0$ | 0.97 | 0.97 | 0.98 | 0.91 | - | - |
| $EA_0$ | 0.97 | 0.97 | 0.98 | 0.91 | - | - |

**Table 2.** Significance of path coefficients.

| Path | Total Indirect Effect | Total Effect | Path | Specific Indirect Effect |
|---|---|---|---|---|
| $EA_0 \rightarrow PB_0$ | | 0.54 *** | $EA_0 \rightarrow PL_0 \rightarrow WTR_0$ | 0.12 *** |
| $EA_0 \rightarrow PL_0$ | | −0.45 *** | $GP_0 \rightarrow PL_0 \rightarrow WTR_0$ | 0.21 *** |
| $EA_0 \rightarrow WTR_0$ | 0.26 *** | 0.15 *** | $EA_0 \rightarrow PB_0 \rightarrow WTR_0$ | 0.20 *** |
| $GP_0 \rightarrow PB_0$ | | 0.36 *** | $GP_0 \rightarrow PB_0 \rightarrow WTR_0$ | 0.08 *** |
| $GP_0 \rightarrow PL_0$ | | −0.46 *** | | |
| $GP_0 \rightarrow WTR_0$ | 0.19 *** | 0.23 *** | | |
| $PB_0 \rightarrow WTR_0$ | | 0.21 *** | | |
| $PL_0 \rightarrow WTR_0$ | | −0.45 *** | | |

Note: *** $p < 1\%$.

**Table 3.** The moderating effects of $P_A$, $P_I$, $P_E$, $P_R$, $P_{RE}$, and $P_O$.

| Path | $P_A$ | | | $P_I$ | | | $P_E$ | | |
|---|---|---|---|---|---|---|---|---|---|
| | $P_{A,O}$ | $P_{A,Y}$ | Δ | $P_{I,H}$ | $P_{I,L}$ | Δ | $P_{E,H}$ | $P_{E,L}$ | Δ |
| $EA_0 \rightarrow WTR_0$ | 0.13 *** | 0.26 *** | −0.14 * | 0.18 *** | 0.16 *** | 0.02 | 0.13 ** | 0.25 *** | −0.13 |
| $GP_0 \rightarrow PB_0$ | 0.47 *** | 0.22 *** | 0.25 ** | 0.44 *** | 0.29 *** | 0.14 | 0.52 *** | 0.25 *** | 0.28 *** |
| $PL_0 \rightarrow WTR_0$ | −0.40 *** | −0.49 *** | 0.08 | −0.34 *** | −0.49 *** | 0.15 *** | −0.46 *** | −0.47 *** | 0.01 |
| $GP_0 \rightarrow PB_0 \rightarrow WTR_0$ | 0.12 *** | 0.03 * | 0.09 *** | 0.13 *** | 0.06 *** | 0.07* | 0.12 *** | 0.03 ** | 0.09 *** |

| Path | $P_R$ | | | $P_{RE}$ | | | $P_O$ | | |
|---|---|---|---|---|---|---|---|---|---|
| | $P_{R,U}$ | $P_{R,R}$ | Δ | $P_{RE,Y}$ | $P_{RE,N}$ | Δ | $P_{O,Y}$ | $P_{O,N}$ | Δ |
| $EA_0 \rightarrow PL_0$ | −0.41 *** | −0.47 *** | 0.06 | −0.41 *** | −0.32 *** | −0.09 | −0.44 *** | −0.73 *** | 0.28 ** |
| $EA_0 \rightarrow WTR_0$ | 0.19*** | 0.08 | 0.12 * | 0.13 ** | 0.24 *** | −0.12 | 0.19 *** | −0.01 | 0.20 * |
| $GP_0 \rightarrow PB_0$ | 0.33 *** | 0.41 *** | −0.08 | 0.51 *** | 0.28 *** | 0.23 ** | 0.37 *** | 0.23 | 0.15 |
| $GP_0 \rightarrow PL_0$ | −0.50 *** | −0.43 *** | −0.07 | −0.45 *** | −0.42 *** | −0.03 | −0.49 *** | −0.12 | −0.36 ** |
| $GP_0 \rightarrow WTR_0$ | 0.26 *** | 0.16 *** | 0.10 * | 0.24 *** | 0.20 *** | 0.05 | 0.21 *** | 0.34 *** | −0.16 |
| $PL_0 \rightarrow WTR_0$ | −0.40 *** | −0.56 *** | 0.19 *** | −0.42 *** | −0.47 *** | 0.06 | −0.43 *** | −0.44 *** | −0.00 |
| $EA_0 \rightarrow PL_0 \rightarrow WTR_0$ | 0.16 *** | 0.28 *** | −0.11 ** | 0.17 *** | 0.15 *** | 0.02 | 0.19 *** | 0.32 *** | −0.12 |
| $GP_0 \rightarrow PL_0 \rightarrow WTR_0$ | 0.20 *** | 0.26 *** | −0.06 | 0.19 *** | 0.20 *** | −0.01 | 0.21 *** | 0.06 | 0.15 ** |
| $GP_0 \rightarrow PB_0 \rightarrow WTR_0$ | 0.06 *** | 0.09 *** | −0.03 | 0.13 *** | 0.05 *** | 0.08 * | 0.07 *** | 0.07 | 0.01 |

Note: *** $p < 1\%$, ** $p < 5\%$, * $p < 10\%$.

Furthermore, consumers' age, income, and education had no significant moderating influence on $EA_0 \rightarrow PL_0$, $GP_0 \rightarrow PL_0$, $GP_0 \rightarrow WTR_0$, $EA_0 \rightarrow PL_0 \rightarrow WTR_0$, and $GP_0 \rightarrow PL_0 \rightarrow WTR_0$, indicating that these effects have no significant difference based on consumers' age, income, and education. The next sections concentrate on the examination of the key moderating effects, which are as follows.

(1) Consumer age has a significant moderating influence on $GP_0 \rightarrow PB_0$ and $GP_0 \rightarrow PB_0 \rightarrow WTR_0$, $EA_0 \rightarrow WTR_0$. The older group ($P_{A,O}$), which is older than or equal to 30, is more sensitive to $GP_0$ than the younger group ($P_{A,Y}$), which is younger than 30. Elders have more life experience and can better grasp government policy and its implications. As a result, given $GP_0$, $P_{A,O}$ will have a higher perception of $PB_0$ and therefore a higher WTR. Individuals, on the other hand, are unfamiliar with WBEV recycling, and younger people are more open to new experiences. As a result, given $EA_0$, $P_{A,Y}$ is more ready to recycle.

(2) The income of consumers has a significant moderating effect on $PL_0 \rightarrow WTR_0$, $GP_0 \rightarrow PB_0 \rightarrow WTR_0$. The path coefficient of $PL_0 \rightarrow WTR_0$ is greater for the low-income group ($P_{I,L}$), whose annual household income is less than or equal to RMB 200,000, than for the high-income group ($P_{I,H}$), whose annual household income is more than RMB 200,000. This demonstrates that the low-income group is more susceptible to $PL_0$. WBEV recycling is typically associated with WBEV replacement, which entails monetary and time expenditures [12].

(3) The education of consumers has a significant moderating effect on $GP_0 \to PB_0$ and $GP_0 \to PB_0 \to WTR_0$. Consumers with a bachelor's degree or above are classed as having higher education ($P_{E,H}$), and the others are categorized as having lower education ($P_{E,L}$). $P_{E,H}$ may have a stronger belief in the efficacy of government policy than $P_{E,L}$ and may have more ability to forecast the probable effects of government policy. Furthermore, $P_{E,H}$ are better learners and may be more knowledgeable about battery recycling. As a result, given $GP_0$, $P_{E,H}$ has a greater $PB_0$, resulting in a higher WTR.

(4) The residence of consumers has a significant moderating effect on $GP_0 \to WTR_0$, $EA_0 \to WTR_0$, $EA_0 \to PL_0 \to WTR_0$, $PL_0 \to WTR_0$. Consumers living in the urban region are defined as $P_{R,U}$, whereas consumers living in the rural region are defined as $P_{R,R}$. $P_{R,U}$ is more susceptible to $GP_0$ and $EA_0$. WBEV recycling infrastructure is often placed in cities, providing urban dwellers with more recycling convenience. As a result, given $GP_0$ or $EA_0$, $P_{R,U}$ are more ready to recycle. $P_{R,R}$ is, however, more sensitive to $PL_0$. Given $PL_0$, $P_{R,R}$ are more averse to recycling, because they are more worried about monetary and non-monetary loss, resulting in $P_{R,R}$ having a greater influence on $EA_0 \to PL_0 \to WTR_0$ than $P_{R,U}$.

(5) The recycling experiences of consumers have a significant moderating effect on $GP_0 \to PB_0$ and $GP_0 \to PB_0 \to WTR_0$. $P_{RE,Y}$ represents the group of consumers who have recycled more than once for other waste, whereas $P_{RE,N}$ represents the unexperienced group. $P_{RE,Y}$ are generally more aware of the government's recycling plans and goals. The limits and incentives of government recycling rules, as well as the perceived benefits of recycling, push them to recycle additional products. Further, their successful recycling experience boosts their $GP_0$ and $PB_0$. Recycling experiences with other products make it easier for $P_{RE,Y}$ to recycle WBEVs. As a result, given $GP_0$, $P_{RE,Y}$ would have greater $PB_0$ and hence more WTR than $P_{RE,N}$.

(6) The ownership of EVs by consumers has a strong moderating influence on $EA_0 \to PL_0$, $EA_0 \to WTR_0$, $GP_0 \to PL_0$, as well as $GP_0 \to PL_0 \to WTR_0$. $P_{O,Y}$ represents the group of EV owners, whereas $P_{O,N}$ represents the group of non-owners. Given $EA_0$, the findings suggest that $P_{O,N}$ would have a smaller $PL_0$. The reason for this might be that $P_{O,N}$ do not own EVs, and so they do not have the same recycling challenges as $P_{O,Y}$. $P_{O,Y}$ has greater experience with EVs and batteries and hence understands WBEVs and the dangers of not recycling. As a result, given $EA_0$, $P_{O,Y}$ has a greater WTR.

## 4. Discussion

Table 4 shows a number of findings. $H_1$ was well supported: $GP_0$ and $EA_0$ both had large direct impacts on $WTR_0$. The model also supports hypothesis $H_2$: $PB_0$ and $PL_0$ had large mediating effects on $GP_0 \to WTR_0$ and $EA_0 \to WTR_0$. Furthermore, $H_3$ was partially supported. The moderating effects of socioeconomic characteristics were also investigated. Gender did not have a significant moderating influence, although other characteristics such as age, income, education, residence, recycling experience, and ownership did.

**Table 4.** Hypothesis and results.

| Path | $H_1$ | $H_2$ | $H_3$ | | | | | | |
|---|---|---|---|---|---|---|---|---|---|
| | | | $P_A$ | $P_G$ | $P_I$ | $P_E$ | $P_R$ | $P_{RE}$ | $P_O$ |
| $GP_0 \to WTR_0$ | Yes | | - | - | - | - | Yes | - | - |
| $EA_0 \to WTR_0$ | Yes | | Yes | - | - | - | Yes | - | Yes |
| $EA_0 \to PB_0$ | | Yes | - | - | - | - | - | - | - |
| $EA_0 \to PL_0$ | | Yes | - | - | - | - | - | - | Yes |
| $GP_0 \to PB_0$ | | Yes | Yes | - | - | Yes | - | Yes | - |
| $GP_0 \to PL_0$ | | Yes | - | - | - | - | - | - | Yes |
| $PB_0 \to WTR_0$ | | Yes | - | - | - | - | - | - | - |

**Table 4.** *Cont.*

| Path | $H_1$ | $H_2$ | $H_3$ | | | | | | |
|------|-------|-------|-------|-------|-------|-------|-------|-------|-------|
| | | | $P_A$ | $P_G$ | $P_I$ | $P_E$ | $P_R$ | $P_{RE}$ | $P_O$ |
| $PL_0 \rightarrow WTR_0$ | | Yes | - | - | Yes | - | Yes | - | - |
| $EA_0 \rightarrow PL_0 \rightarrow WTR_0$ | | Yes | - | - | - | - | Yes | - | - |
| $GP_0 \rightarrow PL_0 \rightarrow WTR_0$ | | Yes | - | - | - | - | - | - | Yes |
| $EA_0 \rightarrow PB_0 \rightarrow WTR_0$ | | Yes | - | - | - | - | - | - | - |
| $GP_0 \rightarrow PB_0 \rightarrow WTR_0$ | | Yes | Yes | - | Yes | Yes | - | Yes | - |

WBEV recycling and repurposing are advantageous for fully using the value of the whole life cycle of EV batteries, as well as for developing the closed loop of environmental protection and promoting the growth of the circular economy. The circular economy is distinguished by resource conservation and recycling, which is important in addressing resource scarcity and environmental challenges. Consumers are significant stakeholders in the power battery recycling cycle, and this paper's examination of their WTR has important policy implications, which are as follows.

First, consumer recycling rules for WBEVs must be developed. Consumers who have a higher favorable opinion of policy efficacy, as found by Bruno et al. [81] and Nguyen et al. [40], are more likely to recycle. However, there is a global dearth of effective policies. According to King and Boxall [82], Australia's waste battery recycling policy and regulation are either immature or nonexistent. WBEV recycling is not regulated in Korea or India, according to Yong and Rhee [83] and Deshwal et al. [84]. Furthermore, China has no specified WBEV recycling policies for consumers [12]. Policy education, in addition to improving policies and regulations, should be conducted to improve consumers' understanding of policy and their impressions of policy efficacy.

Second, consumers' environmental attitudes must be improved. According to the findings, environmental views favorably impact consumers' WTR, which is consistent with previous research [40,72]. As a result, environmental education should be pushed at all levels of schooling to provide inhabitants with information on WBEVs and suitable recycling practices. Furthermore, advertising initiatives that demonstrate the consequences of incorrect WBEV disposal should be launched to improve consumer knowledge of the risk of non-recycling.

Third, big data should be used to help design the recycling system. In marketing theory, understanding consumer behavior is critical. This study has found that the perceived benefits of recycling increase consumers' WTR, but the perceived loss has substantial negative effects, which is consistent with previous research [40,69,81,85–87]. Based on consumer behavior analysis, recycling systems and incentive measures should be carefully developed to increase consumers' sense of benefits while decreasing their impression of loss. Furthermore, different trading modalities, such as battery leasing programs and deposit return arrangements, should be investigated further to encourage consumers to recycle WBEVs.

Fourth, an intelligent recycling platform should be established. The digital recycling platform should include real-time online battery evaluation, valuation, recycling reservation, consultation, and logistics tracking, as well as an option for complaints and suggestions to improve recycling efficiency and consumer satisfaction. Smart logistics and cross-regional recycling collaboration should be created as well.

Fifth, citizens take advantage of the battery refill mode. The EV battery's energy replenishment mechanism comprises charging and refill modes. When compared to the charging mode, the battery refill mode takes the least amount of time. The fast-charging mode of public charging stations takes 30 to 60 min, whereas the slow-charging option takes 6 to 10 h. In comparison, the battery refill mode takes only 3 to 5 min. Currently, the battery refill mode is only utilized in business applications such as taxis and ride-hailing services. If the battery's refill mode were extensively promoted, an EV battery that fulfils the retirement standard may be recycled immediately during the battery's refill mode

procedure, eliminating non-recycling or informal recycling of WBEVs. However, the design of the battery refill mode significantly limits this option.

## 5. Conclusions

The rapid expansion of the EV sector will certainly result in a large number of retired power batteries, which will pose a serious danger to the environment if they are not properly recycled. According to the PLS-SEM model, the perception of government policy efficacy and environmental attitudes have considerable positive direct effects on consumers' recycling intentions. According to the moderating impact model, age, household income, education, residence, recycling experience, and ownership all have moderating effects. Meanwhile, the mediating effects of perceived benefits and perceived loss have been established.

Despite the hopeful findings, there are certain limitations in this work that need be addressed in future research. To start with, the majority of the consumers had no prior experience with recycling WBEVs, and their assessment of the perceived benefits and losses depended heavily on their subjective imagination. However, when more automobiles reach the end of their useful lives, more consumers will be confronted with the subject of recycling, and their judgement may be influenced, leading to a discrepancy between reality and the conclusions of the study. Second, the creation of a WBEV recycling strategy may restructure the recycling system, resulting in recycling structures and procedures that differ greatly from those which exist now, perhaps altering consumers' WTR dramatically. Third, technological advancements may change the EV battery construction concept, resulting in radically new recycling methods, and the relationship between customers' WTR and the reasons examined in this research should be revisited.

Future research might investigate the causative and mediated relationships with various nations in order to investigate the influence of regional heterogeneity. This research has concentrated on Chinese consumers. However, regional differences in consumer psychology and behavior have been thoroughly discussed in the literature on home plastic recycling [88], factor energy efficiency [89], EV uptake, and the reduction of emissions [90,91]. Future research might investigate the causative and mediating impacts of the factors addressed in this paper by taking into account the heterogeneity of government policies, which can be characterized as incentive, regulatory, or educational [92].

Future research might also look into how the variables described in this article influence consumers' decisions to use official or informal recycling routes. Electric waste [93–99], home and municipal solid trash [89,98,100,101], and plastic garbage [102,103] have all been extensively explored. However, scant research has differentiated between official and informal recycling pathways for WBEVs [4,20,104–106].

Further, future research might concentrate on the cost–benefit analysis of the entire WBEV removal, collection, shipping, and remanufacturing process (repurposing or recycling). It might assess the health and safety concerns of WBEVs in the case that they are disposed of, stockpiled, landfilled, reused, remanufactured, repurposed, or recycled.

**Author Contributions:** Conceptualization, methodology, software, validation, formal analysis, investigation, resources, data curation, writing—original draft preparation, M.G. and W.H.; Writing—review and editing, W.H. and M.G.; Supervision, W.H.; Project administration, M.G.; Funding acquisition, M.G. All authors have read and agreed to the published version of the manuscript.

**Funding:** This research was supported by the Wenzhou Association for Science and Technology (grant no. jczc28).

**Institutional Review Board Statement:** Not applicable.

**Informed Consent Statement:** Informed consent was obtained from all subjects involved in the study.

**Data Availability Statement:** Not applicable.

**Conflicts of Interest:** The authors declare no conflict of interest.

**Abbreviations**

| | |
|---|---|
| EVs | Electric vehicles |
| EV | Electric vehicle |
| EOL | End-of-life |
| WBEVs | Wasted batteries of electric vehicles |
| WBEV | Wasted battery of electric vehicle |
| EPR | Extended producer responsibility |
| CNKI | China National Knowledge Infrastructure |
| WOS | Web of Science |
| 4S stores | Automobile sales service stores |
| TPB | Theory of planned behavior |
| GP | Perception of government policy efficacy |
| EA | Environment attitude |
| PB | Perception of benefit |
| PL | Perception of loss |
| RI | Recycling intention |
| PLS-SEM | Partial least squares structural equation modeling |
| SEM | Structural equation modeling |
| CB-SEM | Covariance-based structural equation modeling |

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
