# Peer review of "Consumer Willingness to Recycle The Wasted Batteries of Electric Vehicles in the Era of Circular Economy"

_sustainability, doi:10.3390/su15032630_

Round 1
Reviewer 1 Report (Previous Reviewer 2)
English language - minor spell check required
Author Response
Point 1. English language - minor spell check required.
Response 1:
Thank you very much for the reminding. We have carefully proofread the manuscript. Besides, we have reviewed the manuscript word by word and modified the improper expressions.
Reviewer 2 Report (Previous Reviewer 3)
The manuscript “Consumer’s Willingness to Recycle for The Wasted Batteries of Electric Vehicle in the Era of Circular Economy” is well organized and has an interesting topic. Please find my comments below to further improve the quality of the submitted manuscript.
1. The first sentence of the abstract is not appropriate. Electric vehicles are mainly known for improving the environment, not solving energy concerns (although both are correct). So, I suggest using a more general sentence at the beginning of the abstract.
2. “oil vehicle” is not a common term to be used. Vehicles using oil are mainly known as “internal combustion engine vehicles”.
3. The manuscript should be carefully proofread.
4. To strengthen the background on waste management in the circular economy, please refer to the article “Two decades of research on waste management in the circular economy: Insights from bibliometric, text mining, and content analyses” and cite it properly.
5. In Figure 1-2, please use the same font used in Figure 1.1.
6. The visualization of figure 3 should be improved.
7. There is a table at the end of page 12 and the beginning of page 13 that has no information inside. I guess there was a problem while converting the word file into pdf. Please check this issue.
8. The discussion section should be separate from the results section. Therefore, I suggest removing the headings “4.1. Results of Hypotheses” and “4.2. Policy Implications” and keeping their content in the discussion section. Besides, limitations should be mentioned in the conclusion section, therefore, please transfer the whole section 4.3 to the conclusion section.
Author Response
Response to Reviewer Comments
Point 1. The first sentence of the abstract is not appropriate. Electric vehicles are mainly known for improving the environment, not solving energy concerns (although both are correct). So, I suggest using a more general sentence at the beginning of the abstract.
Response 1:
Thank you very much for indicating the first sentence of the abstract. The original first sentence of the abstract is “Electric vehicles (EVs) are increasingly being utilized to alleviate energy problems and establish sustainable transportation systems”, as suggested, we have replaced it with a more general sentence as “Electric vehicles (EVs) are increasingly being used to benefit the environment and foster the development of a low-carbon circular economy. However, as compared to internal combustion engine cars, spent EV batteries (WBEVs) constitute a different waste, and their recycling mechanism is still in its early stages. WBEV consumer willingness to recycle should be a problem in a circular economy for EV users to be WBEV recycling pioneers.”
Point 2. “oil vehicle” is not a common term to be used. Vehicles using oil are mainly known as “internal combustion engine vehicles”.
Response 2:
Thank you very much for indicating the improper use of the term “oil vehicle”. We have modified it to be “internal combustion engine vehicles” as suggested.
Point 3. The manuscript should be carefully proofread.
Response 3:
Thank you very much for the reminding. We have carefully proofread the manuscript. Besides, we have reviewed the manuscript word by word and modified the improper expressions.
Point 4. To strengthen the background on waste management in the circular economy, please refer to the article “Two decades of research on waste management in the circular economy: Insights from bibliometric, text mining, and content analyses” and cite it properly.
Response 4:
Thank you very much for indicating the background on waste management in the circular economy. We have carefully read the suggested reference article, and cited it properly in the paper to strengthen the background on waste management in the circular economy. Specifically, in page 3, we added the following descriptions:
“Most governments and economies, including China and the United States, have proposed the circular economy idea as a fundamental premise for industrial development in recent years [16-17]. The goal of the circular economy is to keep resources in the economy for as long as possible through recycling or reusing garbage and its byproducts, as well as prolonging the serve of products [18]. Without appropriate waste management, the circular economy cannot be realised [19]. According to Ranjbari et al. [19], substantial research has been conducted on waste management within the circular economy domain during the previous two decades, with 962 relevant publications published in over 200 journals.
These articles were categorised into seven major themes: circular economy transition, environ-mental impacts and lifecycle assessment, bio-based waste management, electronic waste, municipal solid waste, plastic waste, and building and demolition waste. However, the management of WBEVs is omitted, showing a lack of study on WBEV management in the context of the circular economy. Re-cycling is a component of waste management, and this study investigates customers' recycling intentions for WBEVs, contributing theoretically and practically to waste management and the creation of a circular economy in the following areas: (1) provides an analytical framework for the study of consumers' recycling intention for WBEVs; (2) clarifies and discusses the influencing factors of consumers' WBEVs recycling behavior from the three perspectives of causality effect, mediating effect, and mediation effect; and (3) proposes incentive measures and policies for government and recyclers to raise consumers' recycling intention for WBEVs, based on the key influencing factors.”
Point 5. In Figure 1-2, please use the same font used in Figure 1-1.
Response 5:
Thank you very much for indicating the inconsistent font in Figure 1. We have modified the font in Figure 1-2 to make it consistent with Figure 1-1. Besides, all the figures and tables have also been carefully examined.
Point 6. There is a table at the end of page 12 and the beginning of page 13 that has no information inside. I guess there was a problem while converting the word file into pdf. Please check this issue.
Response 6:
Thank you very much for indicating the table at the end of page 12 and the beginning of page 13 that has no information inside. We have checked it and made the necessary modification.
Point 7. The discussion section should be separate from the results section. Therefore, I suggest removing the headings “4.1. Results of Hypotheses” and “4.2. Policy Implications” and keeping their content in the discussion section. Besides, limitations should be mentioned in the conclusion section, therefore, please transfer the whole section 4.3 to the conclusion section.
Response 7:
Thank you very much for your suggestion. We have removed the headings “4.1. Results of Hypotheses” and “4.2. Policy Implications” and kept their content in the discussion section as suggested. Besides, the whole section 4.3 has been transferred to the conclusion section as recommended.
Point 8. The visualization of figure 3 should be improved.
Response 8:
Thank you very much for indicating the visualization of figure 3. We have carefully improved the Figure 3 as bellow:

This manuscript is a resubmission of an earlier submission. The following is a list of the peer review reports and author responses from that submission.
Round 1
Reviewer 1 Report
Thank you for the opportunity to do a review of your article, below is a list of comments that I think should be taken into account to improve the publication, which sadly is not suitable for publication in its current form, and the following errors make it impossible to do a full inspection of the work:
- the abbreviation EVWB used in the paper is not a colloquial abbreviation, since it refers to the term Wasted Batteries of Electric Vehicle please change it to WBoEV or WBEV according to the order of the original notation used both in the title of the paper and in the following sections
- please indicate other keywords that do not overlap with the title of the work
- The paper has no literature references, this is unacceptable, in the current form I can not check the correctness of the citation! Please apply the guidelines of the journal!
- posted graphics have no source
- graphic 2 is unreadable in the created pdf file, spread over two pages 5 and 6
- in the tables used references in the form of *** are illegible, suggests using a color legend
- please add a list of abbreviations and designations
- lack of announcements in the text required by the publisher, for example, the division of labor among the authors
Reviewer 2 Report
The aim of the article is to establish an analysis model for
consumers’ willingness to recycle for EVWB, based on circular economy and consumers’ welfare. The research design, questions, hypotheses and methods were clearly stated. Arguments and discussion of findings were coherent.
I recommend expanding subchapter 2.3. Figure 2 is visually inappropriate, it can be seen disintegrated in the study, this needs to be improved. I would like to further expand the limits of the research!
Reviewer 3 Report
The title of the manuscript (Consumer’s Willingness to Recycle for The Wasted Batteries of Electric Vehicle in the Era of Circular Economy) was at the same time interesting and strange for me. So, I decided to go through it carefully. According to what authors have presented, “This paper aims to establish an analysis model for consumers’ willingness to recycle for EVWB, based on circular economy and consumers’ welfare, to explore the consumer incentives for the construction of EVWB recycling system”. However, it should be noted that batteries of electric vehicles are considered hazardous materials and cannot be easily recycled. Besides, it is far different with municipal solid waste. Therefore, consumers cannot be involved in recycling this kind of waste; and this process should be handled by car manufacturers or other players in the supply chain, not consumers. So, exploring the consumer incentives for the construction of EVWB recycling system makes no sense. This is also confirmed by the authors through their literature review where they have said “Most related literatures mainly focus on the analysis of automobile manufacturers, battery production enterprises, automobile retailers and recycling enterprise but ignores the role of consumers and rarely discuss measures that motivate consumers to participate in EVWB recycling (Dong & Ge, 2022)”. Although the statistical analysis may have been done with details and follow a logical trend, as the manuscript has considered an incorrect topic that can be misleading in the literature, I regret to say that I cannot recommend this paper for publication.